# Approaches to Dose Finding in Neonates, Illustrating the Variability between Neonatal Drug Development Programs

**DOI:** 10.3390/pharmaceutics12070685

**Published:** 2020-07-20

**Authors:** John N. Van den Anker, Susan McCune, Pieter Annaert, Gerri R. Baer, Yeruk Mulugeta, Ramy Abdelrahman, Kunyi Wu, Kevin M. Krudys, Jeffrey Fisher, William Slikker, Connie Chen, Gilbert J. Burckart, Karel Allegaert

**Affiliations:** 1Division of Clinical Pharmacology, Children’s National Hospital, Washington, DC 20010, USA; JVandena@childrensnational.org; 2Pediatric Pharmacology and Pharmacometrics Research Center, University Children’s Hospital Basel, University of Basel, 4056 Basel, Switzerland; 3Intensive Care and Department of Pediatric Surgery, Erasmus MC/Sophia Children’s Hospital, 3000 CA Rotterdam, The Netherlands; 4Office of Pediatric Therapeutics, Office of Clinical Policy and Programs, Office of the Commissioner, U.S. Food and Drug Administration, Silver Spring, MD 20993, USA; Susan.McCune@fda.hhs.gov (S.M.); Gerri.Baer@fda.hhs.gov (G.R.B.); 5Department of Pharmaceutical and Pharmacological Sciences, KU Leuven, 3000 Leuven, Belgium; pieter.annaert@kuleuven.be; 6Division of Pediatric and Maternal Health, Office of New Drugs, Center for Drug Evaluation and Research, U.S. Food and Drug Administration, Silver Spring, MD 20993, USA; Yeruk.Mulugeta@fda.hhs.gov (Y.M.); Ramy.Abdelrahman@fda.hhs.gov (R.A.); 7Office of Clinical Pharmacology, U.S. Food and Drug Administration, Silver Spring, MD 20993 USA; Kunyi.Wu@fda.hhs.gov (K.W.); gilbert.burckart@fda.hhs.gov (G.J.B.); 8Office of Neurosciences, Office of New Drugs, Center for Drug Evaluation and Research, U.S. Food and Drug Administration, Silver Spring, MD 20993, USA; Kevin.Krudys@fda.hhs.gov; 9National Center for Toxicological Research, U.S. Food and Drug Administration, Jefferson, AR 72079, USA; Jeffrey.Fisher@fda.hhs.gov (J.F.); William.Slikker@fda.hhs.gov (W.S.); 10Health and Environmental Sciences Institute, Washington, DC 20005, USA; cchen@hesiglobal.org; 11Department of Development and Regeneration, KU Leuven, 3000 Leuven, Belgium; 12Department of Clinical Pharmacy, Erasmus MC Rotterdam, 3000 CA, The Netherlands

**Keywords:** drug development, newborn, neonatal clinical pharmacology, dose finding, pediatric drug delivery

## Abstract

Drug dosing in neonates should be based on integrated knowledge concerning the disease to be treated, the physiological characteristics of the neonate, and the pharmacokinetics (PK) and pharmacodynamics (PD) of a given drug. It is critically important that all sources of information be leveraged to optimize dose selection for neonates. Sources may include data from adult studies, pediatric studies, non-clinical (juvenile) animal models, in vitro studies, and in silico models. Depending on the drug development program, each of these modalities could be used to varying degrees and with varying levels of confidence to guide dosing. This paper aims to illustrate the variability between neonatal drug development programs for neonatal diseases that are similar to those seen in other populations (meropenem), neonatal diseases related but not similar to pediatric or adult populations (clopidogrel, thyroid hormone), and diseases unique to neonates (caffeine, surfactant). Extrapolation of efficacy from older children or adults to neonates is infrequently used. Even if a disease process is similar between neonates and children or adults, such as with anti-infectives, additional dosing and safety information will be necessary for labeling, recognizing that dosing in neonates is confounded by maturational PK in addition to body size.

## 1. Introduction

Health-care providers still frequently use dosing regimens and drug formulations that initially were developed for adults or children in neonates. As a consequence, dosing regimens in neonates have been based on trial and error from non-neonatal indications primarily studied in adult medicine and are purely based on the pathophysiology in adults [1,2]. In addition to dosing, formulations or dosage forms should be tailored to deliver accurate and safe doses in (pre)-term neonates as their specific needs have recently been summarized [3].

Legal initiatives were taken to improve this setting. In the U.S., the incentives of the Best Pharmaceuticals for Children Act (BPCA) in 2002 and the requirements of the Pediatric Research Equity Act (PREA) in 2003 provided a regulatory framework for the study and approval of new drugs in relevant pediatric populations [4,5]. These Acts were made permanent under the 2012 FDA Safety and Innovation Act (FDASIA). This Act further focused on one of the neglected pediatric groups in drug development, namely neonates [6,7]. In Europe, neonates were included at the initiation of the Pediatric Regulation in 2007 and an EMA Guideline on the Investigation of Medicinal Products in Term and Preterm Neonates was released the same year [8,9]. These legal initiatives have been critically important as the survival of extremely low birth weight infants continues to increase and most of these infants are exposed to medications that are used “off-label”, meaning that these drugs are not specifically FDA- or EMA approved for use in neonates [10,11].

However, these initiatives do not immediately result in better labeling. This is because many of the drugs that have been studied in neonates under BPCA and PREA have little use in neonates, as they are for adult or non-neonatal pediatric indications. In addition, studies of some drugs that have been used for neonatal conditions failed to show efficacy. It is not clear whether these drugs were indeed not effective in neonates or if there were limitations in trial designs, including whether the optimal dose of the drugs was studied, or efficacy was assessed using validated endpoints [10]. Ku and Smith (2015) delineated the factors that need to be taken into consideration for use in neonatal drug studies, including the unique physiology of neonates and the challenges with neonatal study designs. One specific but pivotal challenge is to determine the dose that will be safe and effective in neonates whose gestational age may vary between 22 and 42 weeks and whose birth weights may range between 500 and 5000 g, with additional differences in postnatal age [12]. In utero ontogeny is disrupted by preterm birth and by postnatal interventions including exposure to a relatively hyperoxic environment (even in room air) with potential major changes in organ perfusion and alterations in nutrient/mineral delivery.

It is critically important that all sources of information leveraged to optimize dose selection for neonates take this prenatal and postnatal physiology and its variability into account [13]. These sources may include data from adult studies, pediatric studies, non-clinical juvenile animal models, in vitro studies, and in silico models. Depending on the drug development program, each of these modalities could be used to varying degrees and with varying levels of confidence.

Models to define a starting dose in neonates must take into account the known information, as well as assumptions for yet unknown information, related to the ontogeny (and its variability) of the absorption, distribution, metabolism, and excretion (ADME) of the drug [14]. When the disease is related but not similar to the disease in adults or older children, the bridging approach for similar disease can be used as supportive information, but additional data from clinical trials will be needed to identify the dose in neonates with appropriate ethical justification. In any case, all maturational ADME processes must be taken into account in this setting as well. When the disease is unique in neonates and the drugs have not been used to study disease in adults or older children, dose finding becomes more complicated. In this case, greater reliance must be placed on non-clinical juvenile animal models and in silico models. Understanding the mechanism of action of the drug in neonates is key in determining the on-target and off-target effects of the drug that are responsible for both efficacy and safety. In silico models depend on thorough understanding of the basic science and molecular biology of the disease, the mechanism of action of the drug, neonatal (patho)-physiology, and the ontogeny of the ADME pathways involved in drug handling.

Clinical pharmacology aims to predict drug-specific effects and side-effects based on pharmacokinetics (PK) and pharmacodynamics (PD). PK estimates the relationship between a drug dose and a concentration-time profile (“*what the body does to the drug*”) at a specific site (e.g., plasma, cerebrospinal fluid). PD estimates the relationship between a drug concentration and effects (“*what the drug does to the body*”), and covers both beneficial effects, as well as side effects. Consequently, drug dosing in young infants should be based on integrated knowledge concerning the specific diseases to be treated, the physiological characteristics of the infant receiving the drug, and the PK and PD parameters of the drug. When we consider the changes and subsequent variability in physiological characteristics of neonates, we should be aware that maturational changes in physiology are most prominent in the neonatal period and early infancy. These maturational physiological changes may be further modulated by applied treatment modalities (e.g., whole body cooling, extracorporeal membrane oxygenation (ECMO) or pharmacotherapy) or underlying pathophysiological processes (e.g., perinatal asphyxia, cardiopathy, sepsis, renal failure, patent ductus arteriosus) [13,15].

The International Neonatal Consortium (INC) with participation from Canada, Europe, Japan, and the United States was launched by the Critical Path Institute in 2015. INC developed a comprehensive white paper to guide neonatal clinical trials of medicines—particularly early phase studies—including the need to identify the correct doses in neonates [2]. In this white paper, the concept of extrapolation for initial dose selection and its limitations in neonates is discussed. In essence, this implies on an extrapolation concept based on biological and pharmacological rationale with quantitative predictions and a subsequent decision tree [2].

The current paper further builds on this prior INC work and the Health and Environmental Sciences Institute (HESI) research initiative to increase the knowledge base in the non-clinical neonatal space to better inform decisions on neonatal drug development programs [16]. We hereby aim to illustrate the variety in neonatal drug development programs (meropenem, caffeine, surfactant, thyroid hormone, and clopidogrel) and different dose selection approaches, based on the ability to leverage existing information as follows:Neonatal disease *similar* to that in adults and/or older pediatric patients where dosing is known for adult and/or older pediatric patients (meropenem), as anti-infective use is focused on the infectious agent and not the host.Neonatal disease *related* but not similar to that in adults and/or older pediatric patients where dosing is known for adult and/or older pediatric patients (clopidogrel, and thyroid hormone).Neonatal disease *unique* to neonates where these drugs are not utilized for these specific diseases in adults and/or older pediatric patients (caffeine, and surfactant).

We hereby describe the drug development program as conducted within the United States and assessed by the FDA. However, when applicable we also refer to European activities, as assessed by the European Medicines Agency (EMA).

## 2. Approaches When Adult and/or Pediatric Dosing is Available and When the Disease is Similar to the Disease in Adults and/or Older Pediatric Patients

Extrapolation of efficacy from adults to pediatric—including neonatal—patients is permitted in drug development in the U.S. and within the European Regulation under certain conditions [9,17]. Antibiotics are the most commonly prescribed drugs in neonates [18]. Off-label use of antibiotics can result in unpredictable responses, related to either unanticipated toxicity or lack of efficacy (e.g., target concentration not reached). Moreover, this may induce resistance following inadequate exposure. The use of chloramphenicol with the associated grey baby syndrome is a historical illustration of potentially relevant toxicity, while inadequate dosing may result in therapeutic failure. Off-label use has the potential to result in inadequate or inaccurate dosing and explains the variations in dosing regimens for antibiotics in both US as well as European neonatal intensive care units [19,20].

### Meropenem

Meropenem is a broad-spectrum carbapenem used to treat intra-abdominal infections, including necrotizing enterocolitis (intra-abdominal infection) in (pre)term neonates. This disease is similar, but not identical to abdominal infections in children or adults. Complicated intra-abdominal infections are likely to be more severe in neonates than in older infants and children, but the pathogens (Gram-negative, Gram-positive, and anaerobic) involved are similar. Meropenem is primarily cleared by renal elimination. Consequently, it is very reasonable to postulate that a dosing regimen will reflect maturational changes in renal elimination [21,22].

Based on this similarity, it is reasonable to assume that the efficacy of antimicrobial agents, like meropenem to treat a complicated intra-abdominal infection, can be fully extrapolated from data in older children and adults (‘extrapolation’). Consequently, drug dosing can be performed by matching drug exposure (‘free’ plasma concentration over time) in neonates to that observed in pivotal drug development studies in adults. Consequently, PK studies were performed in neonates with different gestational and postnatal ages to conduct a robust PK analysis aiming at the same target exposures for meropenem shown to be effective in adults and older children. Based on this PK analysis, the dosage schedule for neonates with varying gestational and postnatal ages was determined (Table 1) [23].

The initial focus with meropenem was on complicated abdominal infections, to collect PK and safety data [24,25,26]. Of the studies that supported labeling, one was an open-label, non-comparative, multicenter, prospective PK, and safety study in infants <91 days of age. In this study, 188 (pre)term infants (28 (range 23–40) weeks GA; 21 (1–92) days PNA) received meropenem at 20 or 30 mg/kg every 8 or 12 h and 780 meropenem plasma concentrations were analyzed. Meropenem clearance was associated with creatinine and postmenstrual age (PMA) (clearance [L/h/kg] = 0.12 × [(0.5/creatinine) × 0.27] × [(PMA)/32.7) × 1.46] [24]. Meropenem concentrations remained >4 μg/mL for 50% and >2 μg/mL for 75% of the dose interval in 96% and 92% of patients. Consequently, the authors concluded that the meropenem dosing applied (20–30 mg/kg, every 8 to 12 h, GA and PNA based) in cases with complicated abdominal infections was on target (Table 1).

These PK data supported (in part) the label inclusion of “*complicated intra-abdominal infections*” in neonates, similar to patients >3 months. In patients >3 months, meropenem is additionally labeled for complicated skin and soft tissue infections and bacterial meningitis, but limited data were available to support these indications in neonates. In a very small study using an opportunistic sampling strategy with 9 cerebrospinal fluid (CSF) samples in 6 patients, CSF penetration of meropenem was estimated [24]. More recently, two large studies on meropenem use in neonates (NeoMero1 and NeoMero2) supported by the European Funded Program 7 reported on plasma and CSF PK of meropenem in neonates and young infants [27,28]. This much larger study collected 78 CSF samples in 56 patients and demonstrated that CSF penetration was low (8%). It also showed that with increased CSF protein levels, the CSF penetration of meropenem was more pronounced.

Compared to the adult label, there is no guidance on how to adapt dosing in children or neonates with renal impairment. Modeling and simulation, as illustrated by the earlier mentioned model approaches on renal clearance and its covariates, can be considered [21,22]. PK studies in specific populations like neonates, infants, or children on hemodialysis or extracorporeal membrane oxygenation may guide clinicians to make individual decisions [29,30].

PK data should be supported by safety in neonates and efficacy in other populations. Assessment of safety and reporting of adverse events are essential to assess potential risks of any intervention in neonates. As recently published, assessment (causality, severity) and reporting of adverse events in itself has issues in neonates throughout all phases of neonatal drug development [31]. The Meropenem Study group reported on meropenem safety in neonates and young infants [25]. Adverse and serious adverse events were common. Adverse events included sepsis (6%), seizures (5%), elevated conjugated bilirubin (5%), and hypokalemia (5%). None were judged by the investigator to be probably or definitely related and 2/34 serious adverse events (fungal sepsis, isolated ileal perforation) were judged to be possibly related to meropenem. Seizures were an adverse event of special interest, since this is explicitly mentioned in the Summary of Product Characteristics (SPC). Clinical seizures were observed in 10 (pre)term neonates, confirmed in 1/6 cases with electroencephalographic readings. Moreover, the predicted meropenem Cmax_ss_ in subjects with seizures did not differ from those without seizures. In another retrospective analysis in 5566 infants treated with meropenem or imipenem/cilastatin, mortality and the combined outcome of death or seizures was lower with meropenem (OR 0.68 and 0.77, respectively) [26].

It remains important to be aware that these studies were not powered as an efficacy trial. Consequently, the evidence to support meropenem labeling in neonates with intra-abdominal infections was based on safety and effectiveness data from adequate studies in adults supplemented by PK and safety data in neonates.

## 3. Approaches When Adult and/or Pediatric Dosing is Available and the Disease is Related but not Similar to the Disease in Adults and/or Older Pediatric Patients

For diseases related to but not similar to adults or older children, additional information can be leveraged from either in vitro (clopidogrel, in vitro platelet response) or in vivo (levothyroxine, juvenile animals) models to guide initial dosing. The history of dose finding as well as the clinical study of clopidogrel in neonates is from an educational standpoint illustrative on how errors in dose determination can occur. To explore the mechanisms related to neonatal congenital hypothyroidism associated impairment, and the impact of substitution, juvenile animal experimental models are supportive.

### 3.1. Clopidogrel

Thromboembolism in neonates is usually a complication of another underlying condition such as sepsis, surgery or cardiac disease (in this population, most commonly congenital heart disease). According to the American College of Chest Physicians (2012) the available evidence supporting any recommendation of antithrombotic therapy in neonates was weak [32]. The traditional prophylactic drug for thromboembolism has been aspirin. The limitation of the use of aspirin is that it only inhibits platelet activation through the cyclooxygenase pathway. Clopidogrel is a thienopyridine derivative prodrug which undergoes conversion to its active metabolite via a two-step mechanism that is mediated by several cytochrome P450 (CYPs), including CYP2C19 and CYP3A4. This active metabolite binds specifically and irreversibly to the ADP purigenic P2Y_12_ platelet receptor. This results in the activation of the IIb/IIIa complex with subsequent inhibition of platelet aggregation. Clopidogrel has been demonstrated to be an effective antithrombotic agent in adults [33,34,35]. In the period 2000–2009, data from the Pediatric Health Information System database suggested that the use of clopidogrel increased significantly in pediatrics as well as in neonates [36].

The history of dose finding as well as the clinical study of clopidogrel in neonates is from an educational standpoint illustrative on how errors in dose determination can occur. It starts in 2001, when a Written Request (WR) for studies with clopidogrel was issued by the FDA in response to a Proposed Pediatric Study Request from the sponsor. In 2007, a final revision of the clopidogrel WR was issued by the FDA.

The Platelet Inhibition in Children on Clopidogrel (PICOLO) Trial (dose-finding study for clopidogrel in neonates) supported by the sponsor was published in 2008 [37]. This study was a prospective, randomized placebo controlled trial of clopidogrel in neonates and infants 0 to 24 months of age. Four doses were chosen, but they varied (by weight) from approximately 1% to 20% of the adult dose. Most patients on clopidogrel were also treated with aspirin, which may have accounted for the reluctance to use higher doses. Oral doses were prepared extemporaneously including up to 1:99 dilutions. Pharmacokinetic samples were obtained for determination of plasma concentrations of the inactive carboxylic acid metabolite (SR26334). Platelet aggregation was assessed at baseline and at “steady state” (7 to 28 days after the start of therapy) using an assay for platelet aggregation (based on the addition of 5 μM ADP) published in 1963. The results of the dose-response were reported in the PICOLO study, but the variability in response complicates the interpretation of dose-response. It was suggested that the extent and rate of platelet aggregation at the dose of 0.2 mg/kg/day was similar to that seen in adults at a dose of 75 mg/day. The authors of the PICOLO study concluded that the 0.2 mg/kg/day clopidogrel was the appropriate dose to take into the larger clinical study, and that neonates were more sensitive to the effects of clopidogrel.

In retrospect, these conclusions were made despite (a) accumulating evidence about clopidogrel dosing in different populations, and (b) evidence from a later study of another P2Y_12_ agent, ticagrelor, that an infant’s in vitro platelet response was similar to older children and adults [38]. Clopidogrel dosing in pediatric patients down to 6 weeks of age had already been reported in 2005 using a more ‘traditional’ trial and error approach that had previously been used with off-label drugs in pediatrics. The use of clopidogrel at Toronto Sick Children’s Hospital in 15 infants and children from 6 weeks to 16 years of age was reported. Doses of clopidogrel varied from 1 to 6 mg/kg/day for a duration of therapy of 1 to 6 months. Fourteen of the 15 pediatric patients had no adverse effects from the clopidogrel, but one child on aspirin, warfarin, and clopidogrel had a massive gastrointestinal bleeding. Bleeding times were not measured in the patients on clopidogrel. These authors concluded that a 1.0 mg/kg/day starting dose for clopidogrel in pediatric patients was appropriate.

As follow-up of the PICOLO study, The Clopidogrel to Lower Arterial Thrombotic Risk in Neonates and Infants Trial (CLARINET) investigated the use of clopidogrel (dose = 0.2 mg/kg/day) compared to standard therapy (aspirin) in infants (n = 906) with cyanotic congenital heart disease and a systemic-to-pulmonary shunt [39]. This was a double-blind placebo controlled, event-driven trial. Four hundred sixty-seven infants were treated with 0.2 mg/kg/day clopidogrel, and 439 infants were treated with placebo. All infants were 92 days of age or less. A composite endpoint of death or heart transplantation, shunt thrombosis, or performance of a cardiac procedure due to a thrombotic event before 120 days of age was used. Aspirin was administered concurrently in most patients. The rate of the composite endpoint did not differ significantly between the clopidogrel group (19.1%) and the placebo group (20.5%) [39].

In their review of the data for a pediatric formulation of clopidogrel, the FDA suggested that the exposures of clopidogrel may have been too low in the CLARINET study. There are several considerations that might explain why the CLARINET trial failed even though the PICOLO study showed a pharmacodynamic effect in neonates that was considered to be similar to adults.

First, an important consideration was not included in the study design when it was finalized in 2007. As earlier mentioned, clopidogrel undergoes a two-step activation process to form its active metabolite. In 2006, Hulot et al. published a study on the critical effect of CYP2C19 polymorphisms on the activity of clopidogrel [40]. Eight of their 28 subjects had a CYP2C19*2 loss-of-function allele which was associated with a marked decrease in platelet responsiveness to clopidogrel. Therefore, CYP2C19 genotyping was thereafter considered critical for any research study of clopidogrel in order to distinguish non-responsiveness due to factors other than loss of activity secondary to drug metabolism. The current product labeling for clopidogrel includes a boxed warning for the low response genotype. However, CYP2C19 genotyping was not incorporated into the infant study of clopidogrel. Second and likely even more relevant to this population, maturation of enzyme systems needs to be considered in the context of drug metabolism, especially in neonates where large changes can occur within a short timeframe. The developmental pharmacogenomics (ontogeny + pharmacogenetics) of CYP2C19 in neonates and young infants has been reported with omeprazole as probe drug more recently. At birth, CYP2C19 expression levels are 20–25% of adults and by 1 year of age this has increased to 40–50% [41,42]. This means that the neonate’s ability to activate clopidogrel is substantially lower than that of adults.

In the FDA’s Clinical Pharmacology review, the assessor stated “The geometric mean SR26334 C_max_ (measurement on Day 1 between 0.17 and 3 h post-dose) from 5 neonatal patients in PICOLO receiving the 0.2 mg/kg dose was 0.03 mg/L. According to the relative bioavailability study, the mean C_max_ of SR26334 following a single 75 mg dose in healthy adult male volunteers ranged from 2.8 to 3.3 mg/L. This difference in C_max_ is remarkable, even after taking into account the small pediatric sample size, wide sampling window for C_max_ and the fact that only the inactive metabolite was measured.” The FDA’s medical officer review stated that “at a minimum, given the uncertainties of dose selection, more than one dose should have been tested in the confirmatory trial.” This conclusion is relevant, and does reflect the need to confirm exposure, and to consider these uncertainties during the study design process.

Third, data in the literature additionally demonstrate that neonates have different baseline platelet aggregation (30–40% lower) compared to adults [43]. In this study platelet aggregation in pediatric patients (neonatal and infant data from the PICOLO study) and adults was examined. The authors concluded that the lower baseline platelet aggregation in neonates and infants justified the lower dose of clopidogrel suggested by the PICOLO authors, as this achieved the same PD effect in adults compared to the estimated neonatal dose based on the adult dose. Given that the baseline responses to ADP induced platelet aggregation are different, it is not unreasonable, however, to expect a 20–50% inhibition of platelet aggregation (similar to that achieved by clopidogrel in adults) to result in different clinical effects in adults and neonates [43]. In addition, given the maturation of CYP2C19, an additional arm with a higher dose in neonates might have been prudent. Additionally, measurement of prodrug levels as well as the active metabolite may have supplemented the dosing and safety information. Subsequent to this study and others in infants and children, dose ranging has been incorporated as routine part of dosage assessment for efficacy studies in pediatric patients.

### 3.2. Thyroid Hormone

Thyrotropin-releasing hormone (TRH) is released from the hypothalamus and stimulates secretion of thyrotropin-stimulating hormone (TSH) from the anterior pituitary. TSH subsequently stimulates the synthesis and secretion of mainly L-thyroxine (T4) by the thyroid gland. After secretion, the majority of T4 (80%) undergoes deiodination to L-triiodothyronine (T3) in peripheral tissues. Circulating T4 and T3 subsequently exert a feedback effect on both TRH and TSH secretion. Thyroid hormones regulate multiple metabolic processes and play an essential role in normal growth and maturation of the central nervous system. Thyroid hormone deficiency during fetal life or during infancy results in growth restriction and developmental retardation. Consequently, rapid restoration of normal thyroid function is essential to prevent these adverse events in congenital hypothyroidism, but overtreatment may also affect the rate of brain and bone maturation (e.g., craniosynostosis, premature closure of the epiphyses). This means that doses should be individualized to response (TSH, T4). Dosing guidance for levothyroxine to newborns, infants, and children is summarized in Table 2.

The recommended dose in newborn infants is 10–15 µg/kg/day. In neonates with very low total T4 concentrations (<5 µg/dL), it is recommended to start with 50 µg/day. Using normalization of TSH as target, this is somewhat higher (µg/kg/day) than the recommendation for older infants or children. In addition, in these cases, levothyroxine is usually initiated at full replacement doses. However, in children with chronic or severe hypothyroidism, an initial dose of 25 µg/day is recommended with 2–4 week increments (25 µg/day) until the desired effect is achieved. Hyperactivity in an older child can be minimized if the starting dose is 25% of the recommended full replacement dose, with a subsequent weekly 25% increase until the full replacement dose is reached [44]. The serum total- or free- T4 should be maintained in the upper half of the normal range at all times. While the aim of therapy is to also normalize the serum TSH level, this is not always possible in a small percentage of patients, particularly in the first few months of therapy.

There is variability (40–80%) in T4 absorption following oral levothyroxine administration. Absorption occurs in the jejunum and upper ileum and is increased by fasting and reduced by co-exposure to compounds such as ferrous sulphate, some calcium containing products, soy bean containing infant formula, or simethicone. Some of these compounds are administered to newborns, and potential effects on absorption should be considered [45,46].

Thyroid hormones are highly protein bound in plasma (thyroxine binding globulin, thyroxine binding pre-albumin, albumin). Differences in affinity of T4 and T3 to these binding proteins in part explain the differences in the metabolic clearance and elimination half-life. Only the unbound thyroid hormones are biologically active, but the overall plasma proteins concentrations display changes associated with ontogeny. For albumin, competitive binding with other compounds including bilirubin has been described [47]. The major pathway of thyroid metabolism is sequential deiodination. About 80% of T4 is converted to yield equal amounts of T3 and reversed T3 (rT3) with subsequent additional deiodination to diiodothyronine. Thyroid hormones and their derivatives are either eliminated by the kidney or are conjugated and subsequently excreted in stool after biliary excretion. The urinary fraction decreases with increasing age.

As a result that rapid restoration of thyroid hormones is warranted in neonates with congenital hypothyroidism, one generally starts with 30–50 µg/day (7–15 µg/kg/day). In the first weeks of treatment, closer monitoring for cardiac overload, arrhythmias, or aspiration (too avid suckling) is recommended. In the subsequent follow up, monitoring of thyroid function and clinical monitoring of neurodevelopment and growth/bone maturation are recommended. Finally, when permanent hypothyroidism has not yet been confirmed, it is recommended to consider a 30 day discontinuation trial only when the child is at least 3 years old.

In addition to congenital hypothyroidism, postnatal thyroid hormones in preterm neonates have also been evaluated to prevent morbidity and mortality and in preterm cases with transient hypothyroxinemia. Based on meta-analysis for both indications (primary or secondary prevention), there is insufficient evidence for any of these indications [48].

Although dose selection was largely informed by data in older infants and children, to explore the mechanisms related to neonatal congenital hypothyroidism associated impairment, and the impact of substitution, juvenile animal experimental models were developed. The hypothalamic-pituitary-thyroid (HPT) axis of pregnant rats and rat pups can be readily altered to cause hypothyroidism by surgery (thyroidectomy), drugs, depletion of dietary iodine, or radiation (e.g., I-131) [49,50,51].

Hypothyroidism during the perinatal period in rats alters brain growth and development, modifies the maturation of axonal networks, and impairs synaptogenesis. Thyroxine has been used to investigate its ameliorating effects on these neurodevelopmental processes and behavior in rats. Pups born to hypothyroid pregnant dams exhibited brain abnormalities that were ameliorated if treated with thyroxine [52]. Additionally, the neonatal rat has been used to evaluate the role of altered systemic thyroxine concentrations in retinopathy [53]. Eyes open once the retinal vasculature is complete. In humans, eyes open at birth, and in mice and rats, near postnatal day (PND) 12. The term birth rodent mimics immature eye development for extreme preterm (22–24 weeks gestational age) human births. Raising and lowering oxygen levels daily in rat pups to induce retinopathy mimics the in utero and ex utero oxygen changes in the premature infant that are associated with retinopathy [54].

In summary, qualitatively, rodents are a useful model to explore mechanistic and molecular information about hypothyroidism and its effects on neurodevelopment, including eye development. Juvenile rodents are also useful to study windows of susceptibility for thyroxine treatment to reduce adverse irreversible neurodevelopmental outcomes. Quantitatively there are differences in the functioning of the HPT axis between rats and humans, but also between adults and young within a species [55]. Age dependent changes occur in the HPT axis in both maturing rats and humans. The immature HPT axis functions at an accelerated rate (on a per kg basis) with high turnover in thyroid hormones in the thyroid gland [56,57]. For example, T4 secretion rates are 1.5 and 10 µg/kg/d in euthyroid adult humans and newborn infants, respectively, and 5 and 17 µg/kg/d for the euthyroid adult rat and PND 14 rat pups, respectively. With quantitative consideration of the species differences in HPT function, laboratory animals provide a valuable tool for mechanistic research [44,55]. This information, in addition to clinical data in older infants and children, can provide a strong pharmacological basis for dose selection in neonates.

## 4. Approaches When Neonatal Disease is Unique to Neonates Where These Drugs Are Not Utilized for These Specific Diseases in Adults and/or Older Pediatric Patients

### 4.1. Caffeine

Apnea of prematurity (AOP) is a self-limited condition in preterm infants, which occurs as a direct consequence of immature respiratory control. Since the early 1970s, the primary pharmacologic agents to treat apnea of prematurity are caffeine and theophylline. Caffeine is the preferred agent because of its longer half-life and larger therapeutic window. Given the extensive experience and body of information already available in the published literature regarding the use of caffeine in preterm infants, a summary of a comprehensive search of the literature was used to support the FDA approval of caffeine citrate for use in the preterm neonate in 1999.

The clinical development program for caffeine citrate included a single double-blind placebo controlled clinical trial in preterm neonates with AOP. Sparse blood samples were collected in the clinical trial and the results of a population PK analysis of caffeine plasma levels were provided to support PK in preterm neonates. However, a dedicated PK or dose ranging study was not conducted in preterm neonates to inform dose selection for the efficacy/safety trial [58]. Literature information on caffeine PK in adults and young infants was used as supportive evidence [58]:

*Absorption*: After oral administration of 10 mg caffeine base/kg to preterm neonates, the peak plasma level (C_max_) for caffeine ranged from 6–10 mg/L and the mean time to reach peak concentration (T_max_) ranged from 30 m to 2 h. The extent of absorption (in terms of C_max_ and AUC_0-12h_) was not affected by formula feeding with a slightly longer mean T max in the formula feeding group as compared to the fasting group. The absolute bioavailability was not fully examined in preterm neonates, although it is expected to be 100% [58].

*Distribution*: Caffeine is rapidly distributed into the brain. Caffeine levels in the cerebrospinal fluid of preterm neonates approximate their plasma levels (mean ratios being 0.9–1.0). The mean volume of distribution of caffeine in infants (0.8–0.9 L/kg) is slightly higher than that in adults (0.6 L/kg). Plasma protein binding data are not available for neonates or infants. In adults, the mean plasma protein binding in vitro is reported to be approximately 36% [58].

*Metabolism*: Hepatic cytochrome P450 1A2 is involved in caffeine biotransformation. However, caffeine metabolism in preterm neonates is limited due to their immature hepatic enzyme systems. Interconversion between caffeine and theophylline has been reported in preterm neonates. Following theophylline 7-methylation, caffeine levels are approximately 25% of theophylline levels after theophylline administration and approximately 3–8% of caffeine administered would be expected to convert to theophylline [58,59].

*Elimination*: In young infants, the elimination of caffeine is much slower than that in adults due to immature hepatic and/or renal function. Mean half-life (T1/2) of caffeine in infants have been shown to be inversely related to gestational/postmenstrual age [59]. In neonates, the T1/2 is approximately 3–4 days. By 9 months of age, the metabolism of caffeine approximates that seen in adults (T1/2 = 5 h). Studies examining the pharmacokinetics of caffeine in neonates with hepatic or renal insufficiency have not been conducted.

As mentioned above, the caffeine citrate, intravenous, and enteral (CAFCIT) development program consisted of a single clinical trial. The trial was a multicenter, randomized, double-blind trial compared caffeine citrate to placebo in 85 preterm infants (gestational age 28 to <33 weeks) with apnea of prematurity. Apnea of prematurity was defined as having at least 6 apnea episodes of greater than 20 s duration in a 24-h period with no other identifiable cause of apnea. A 1 mL/kg (20 mg/kg caffeine citrate providing 10 mg/kg as caffeine base) loading dose of CAFCIT was administered intravenously, followed by a 0.25 mL/kg (5 mg/kg caffeine citrate providing 2.5 mg/kg of caffeine base) daily maintenance dose administered either intravenously or orally (generally through a feeding tube). The duration of treatment in this study was limited to 10 to 12 days. The protocol allowed infants to be “rescued” with open-label caffeine citrate treatment if their apnea remained uncontrolled during the double-blind phase of the trial. The percentage of patients without apnea on day 2 of treatment (24–48 h after the loading dose) was significantly greater with CAFCIT than placebo. Table 3 summarizes the clinically relevant endpoints evaluated in this study [58].

The mean number of days with zero apnea events was 3.0 in the CAFCIT group and 1.2 in the placebo group. The mean number of days with a 50% reduction from baseline in apnea events was 6.8 in the CAFCIT group and 4.6 in the placebo group. Along the same line, the Committee for Medicinal Products for Human Use (CHMP) of EMA adopted a positive opinion and granted marketing authorization for caffeine citrate products (Peyona: 20 mg/mL; Gencebok: 10 mg/mL) for the same indication [59,60,61].

In summary, the clinical development program for caffeine citrate included a single double blinded placebo controlled clinical trial in preterm neonates with AOP with sparse PK sampling. In contrast, in settings where the disease is unique to neonates, a PK or PK/PD study would likely be conducted to inform dose selection for the pivotal efficacy study and evidence of effectiveness from more than one adequate and well controlled trial may be needed if feasible. In this case, the extensive clinical experience with the use of caffeine in infants with AOP was used as supportive evidence. Incorporation of sparse blood samples in the efficacy trial also provided additional evidence for dose confirmation.

### 4.2. Lucinactant (Surfaxin)

Respiratory Distress Syndrome (RDS) is the leading cause of death in preterm neonates. With optimal respiratory support, surfactant replacement therapy (SRT) has significantly increased survival among extremely low birth weight infants at risk for developing RDS. A total of five surfactants have been approved by the FDA since 1990, including beractant (Survanta), calfactant (Infasurf), poractant alpha (Curosurf), colfosceril (Exosurf), and lucinactant (Surfaxin). Among these five surfactants, colfosceril was the first one approved by the Agency in 1990 for prevention and treatment of RDS in premature infants, while lucinactant was the latest one approved by the FDA for the prevention of RDS in premature infants at high risk of RDS (Table 4) [62,63,64,65,66]. Beractant, calfactant, and poractant alpha are from animal origin, while colfoscenril and lucinactant are synthetic or synthetic protein analogs to address concerns over potential immunogenicity and transmission of infectious diseases from the animal-derived products. All surfactants approved in the United States are administrated directly to the lungs through an endotracheal tube by bolus, and have no systemic absorption or effect.

RDS is a clinical condition found almost exclusively in premature infants characterized by inadequate production of endogenous pulmonary surfactant that is required to reduce surface tension at the pulmonary alveolar air/liquid interphase. Consequently, the dose selection of surfactants cannot be directly extrapolated from adults based on a pharmacokinetic exposure match [62,63,64,65,66]. Therefore, evidence from adequate and well-controlled investigations in premature infants is required to support this specific indication. As a result, dose and dosing regimen selection in the clinical studies is an important part of the RDS drug development program. This section will focus on the dose selection approach for lucinactant and the data used to support approval for prevention of RDS in its drug development program.

Lucinactant (Surfaxin) was developed as a synthetic surfactant composed of phospholipids and a high concentration of a synthetic, hydrophobic, 21-amino acid peptide (sinapultide-B SP-B) intended to mimic the structural and functional properties (surface tension lowering) of endogenous SP-B, which appears to play a major role in reducing alveolar surface tension. In vitro studies showed that lucinactant decreased surface tension, inhibited superoxide production and resisted inhibition by serum components. Lucinactant is administered directly to the lung, where biophysical effects occur at the terminal airways and alveolar surface [66]. As a result, no human PK studies have been performed to characterize systemic ADME of this product. Therefore, the NDA (new drug application) submission of lucinactant does not contain any human PK information to facilitate dose or dosing regimen selection. However, the drug activity of lucinactant was also investigated in premature animals and acute RDS (ARDS) animal models. Lucinactant has been shown to effectively increase lung expansion and improve gas exchange in both premature rabbits and monkeys as well as in rabbit and pig ARDS models.

A total of three efficacy and safety studies were conducted in premature animals. An efficacy and safety study was conducted in premature rhesus monkeys delivered by Caesarian section on Days 127–131 gestation age (about 0.8 of gestation). In this study, four premature monkeys each were randomly assigned to two groups and treated with a single dose of either 133 mg/kg (5 mL/kg) of lucinactant or the comparator, colfosceril (Exosurf) intratracheally, and were maintained under mechanical ventilation with monitoring of all vital signs for 10 to 13 h, and then sacrificed. The results indicated that administration of lucinactant increased gas exchange and decreased ventilator pressure in premature monkeys while the monkeys receiving colfosceril showed little or no improvement [63]. Lucinactant and colfoscreril were administered at a mean time of 2.8 h and 2.9 h after birth, respectively. Another open-label efficacy and safety study was conducted with 7 premature monkeys treated with doses 126 mg to 200 mg/kg for longer duration. Based on the results summarized in FDA’s Pharmacology review for the NDA, it seems that the higher dose of lucinactant may achieve a numerically higher arterial to alveolar (a/A) oxygen ratio. It also reported that those premature monkeys who received higher doses of lucinactant responded faster compared to the ones who received lower doses. However, the study did not include untreated cohorts. A pulmonary dynamic compliance study was conducted in premature rabbits (27-day gestation). A dose of 100 mg/kg lucinactant was administered intratracheally to premature rabbits. The results indicated that lucinactant significantly increased compliance of the respiratory system in premature rabbits 20-30 min post dosing compared to the control group (phospholipid only) [61].

Three clinical studies were conducted in premature neonates in the lucinactant drug development program. Two of the studies investigated the prevention of RDS and one study investigated RDS treatment once it had occurred. Of the two prevention efficacy studies, KL4-IRDS-06 was the single pivotal study. The other study provided supportive data. Both studies used the same dose of 175 mg/kg. The initial clinical doses, 133 mg/kg and 200 mg/kg, were selected based on results in monkey studies. A clinical study was designed to compare those two doses, but the extent to which the study contributes information on dose selection for prevention use is unknown, as this study was to treat RDS. In addition, only eight out of thirty-nine patients received the low dose (133 mg/kg) compared to thirty-one patients who received the high dose (200 mg/kg). Therefore, the dose selected (175 mg/kg) for Phase 3 development was arbitrary although it was in the range of doses evaluated in the initial clinical study and animal studies. The pivotal study was a multi-center, randomized double-blind, active-controlled parallel group study conducted in premature neonates between 600 g and 1250 g birth weight. Infants were randomized at a ratio of 2:2:1 to lucinactant (Surfaxin©), colfosceril (Exosurf©), and beractant (Survanta©), respectively. Colfosceril was considered the primary comparator. Patients were stratified within each category by birth weight. The first dose of surfactant was given between 15 and 30 min after birth and up to three subsequent doses could be given at a minimum of 6-h intervals if certain predefined criteria consistent with development of RDS were met. The dose of lucinactant was 175 mg/kg up to maximum of 4 administrations. In addition, since the volume of a single dose (5.8 mL/kg) represents more than 50% of a premature neonate’s tidal volume (8–10 mL/kg), the total volume of a dose is divided into four quarter-dose or aliquots. In this study, lucinactant was superior to the active comparator, colfosceril, on both co-primary endpoints, the incidence of RDS at 24 h and RDS mortality at 14 days. Specifically, the incidence of RDS was about 17% less in patients treated with lucinactant than with the active comparator colfosceril, and RDS-related mortality was approximately half the rate in lucinactant patients (4.7 vs. 9.6%). Results were consistent across population subgroups based on birth weight, gender, and race. Adverse events were consistent across treatment arms.

Lucinactant is a synthetic amino acid residue peptide which may potentially mount an immune response. However and due to the immature immune system in premature neonates along with the negative immunotoxicity study results in guinea pigs, a clinical immunogenicity assessment was not performed. Although clinical studies demonstrated the efficacy of lucinactant, the dose and dosing regimen in the Phase 3 pivotal study were selected empirically but in the range of doses evaluated in one clinical study and animal studies.

In summary, this case exemplifies the complexity of neonatal dose selection when the condition is severe and unique, the product is locally acting, and the target population may have significant comorbidities and different ranges of maturity. In such cases, where relevant data from adults or older children are lacking, data from animal studies, especially a well-designed dose finding study in a relevant animal model, combined with clinical data in the target population can support dosing. In the absence of current studies, immunogenicity assessments in neonates should be performed when appropriate for drugs with the potential for immunogenicity.

## 5. Discussion

Establishing dosing strategies (dose and dosing regimen) can be difficult for neonates. In general, juvenile animal studies are conducted as proof of concept with respect to efficacy of a therapeutic agent, but they also can provide vital information with respect to dosing [16]. It is important that all available information be leveraged to help identify the appropriate neonatal doses. Ontogeny of organs and enzyme systems may influence ADME, all of which may affect decisions on dosing strategies [1,12,14]. The subsequent translation to a specific neonatal drug development program differs, mainly driven by disease similarity, disease relatedness, or a disease unique to neonates.

When the ***disease is similar*** in neonates to the disease in older children and adults, efficacy information for the given drug may be extrapolated and dosing can be targeted by matching the PK/PD characteristics documented in older children and adults to neonates. Understanding the mechanism of action of the therapeutic product and the ontogeny of organ systems can help to predict on-target and off-target effects. This is particularly important with respect to safety evaluations and prediction of drug disposition in specific organs, such as central nervous system penetration in neonates. In addition, PK/PD studies can be used to identify the appropriate dosing strategy and safety studies can focus on potential signals identified in studies of older children and adults.

When the ***disease is related but not similar*** to that seen in older children and adults, it is important to understand where the disease differs. Information that is known, such as developmental pharmacogenomics, should be leveraged in neonates. In addition, it is important to consider the ontogeny of any systems that influence ADME as these may affect any decisions with respect to dosing strategies.

However, ***many of the diseases of neonates do not have correlates in older children and adults***. In these cases, it is critically important to leverage all the available information to identify potential doses to study. Animal models, including those of premature or juvenile animals, may help with proof of concept and suggested dosing. It may be necessary to study additional doses to bracket those suggested by adult and animal studies. Dose ranging studies will be critical and may be empirical in nature. The dose approach can be narrowed by information both from studies in adults and/or children, if the drug has been used in adults, and from premature or juvenile animal models [2,16]. The ideal models should include similar ontogeny of the organ systems being studied.

Regardless of the approach, there is increasing interest in appropriate, in vitro and in silico approaches such as the application of physiologically-based pharmacokinetic (PBPK) modeling to predict dosing in neonates. Basic science studies have recently increased the knowledge of the ontogeny of metabolizing enzyme systems, allowing for better modeling of dosing in neonates. PBPK modeling will be important for neonatal specific diseases but the accuracy of the models will depend on the robustness of the knowledge of the ontogeny of the organ systems being studied and the extent to which the ADME pathways of the therapeutic products have been characterized [67]. For caffeine, PK data as collected during clinical studies in neonates were used to further develop such PBPK models and to evaluate their predictive performance [68]. In contrast, such PBPK models were also reported for meropenem and clopidogrel, but have not yet been extended to neonates [69,70]. This illustrates both the potential, but also the still existing opportunities to leverage these sources of information to further optimize dose selection for neonates.

Finally, the need for dose accuracy in (pre)term neonates necessitates an additional focus on age-appropriate formulations [3]. When applied to some drugs considered in this review, oral doses were prepared extemporaneously including up to 1:99 dilutions for clopidogrel, and the mg/mL dose of surfactant differs between different formulations, while there are currently different caffeine (strength) formulations registered in Europe (10 or 20 mg/mL) [59,60]. The development of improved dosing formulations to provide additional accuracy should be informed by a risk assessment on dosing errors because of bedside unawareness of the different formulations. While the currently available formulations result in rather accurate dosing for levothyroxine in newborns (10–15 µg/kg/day, Table 2), this is much less the case in preterm neonates.

## 6. Conclusions

This is an exciting era where new therapeutic products are being studied for neonatal diseases. Accurate approaches to dose ranging studies must leverage information as available from relevant information in adults and other children, animal models of the specific diseases, basic science knowledge of the ontogeny of organ and enzyme systems, and assays that reflect the normal maturation of neonates. Using specific examples, we have illustrated the feasibility and limitations of such a leverage strategy to neonatal drug development.

## Figures and Tables

**Table 1 pharmaceutics-12-00685-t001:** Recommended meropenem IV dosage schedule for pediatric patients less than 3 months of age with complicated intra-abdominal infections and normal renal function (GA: gestational age (at birth), weeks and PNA, postnatal age, days) [23].

Age Group	Dose (mg/kg)	Dose Interval
infants < 32 weeks GA and PNA < 2 weeks	20	every 12 h
infants < 32 weeks GA and PNA ≥ 2 weeks	20	every 8 h
infants ≥ 32 weeks GA and PNA < 2weeks	20	every 8 h
infants ≥ 32 weeks GA and PNA ≥ 2 weeks	20	every 8 h
*there is no recommendation in pediatric patients with renal impairment*		

**Table 2 pharmaceutics-12-00685-t002:** Levothyroxine dosing guidelines in newborns, infants, and children for pediatric hypothyroidism. Doses should be individualized to patient response based on clinical response and laboratory parameters (thyrotropin-stimulating hormone (TSH), T4).

Age	Dose (µg/kg/day)
0–3 months	10 to 15
3–6 months	8 to 10
6–12 months	6 to 8
1–5 years	5 to 6
6–12 years	4 to 5
>12 years, growth and puberty incomplete	2 to 3
growth and puberty completed	1.7

**Table 3 pharmaceutics-12-00685-t003:** Efficacy analysis in 82/85 preterm cases included in the caffeine citrate, intravenous, and enteral (CAFCIT) trial (3 cases are not included in the efficacy analysis because they had <6 apnea episodes/24 h at baseline) [58].

Study Characteristics	Caffeine	Placebo	p-value
number of cases evaluated	45	37	
number (%) without apnea on day 2	12 (26.7%)	3 (8.1%)	0.03
apnea rate on day 2 (/24 h)	4.9	7.2	0.134
number (%) of cases with 50% reduction in apnea events from baseline to day 2	(76%)	(57%)	0.07

**Table 4 pharmaceutics-12-00685-t004:** Surfactant products approved by the FDA for prevention and/or treatment of Respiratory Distress Syndrome (RDS) (NDA= new drug application).

Product	NDA Number Approval Date	Product Information	Indication, Respiratory Distress
colfosceril [62]	20044 (August 1990)	*Synthetic* colfosceril palmitate 67.5 mg/mL; tyloxapol; cetyl alcohol	prevention treatment
beractant [63]	20032 (July 1991)	*Bovine* Phospholipids 25 mg/mL; SP-B < 0.2 mg/mL	prevention treatment
calfactant [64]	20521 (July 1998)	*Bovine* Phospholipids 25 mg/mL; SP-B 0.26 mg/mL	prevention treatment
poractant-alpha [65]	20744 (November 1999)	*Porcine* phospholipids 80 mg/mL; SP-B 0.3 mg/mL	treatment
lucinactant [66]	21746 (March 2012)	*Synthetic* sinapultide 0.8 mg/mL	prevention

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
