# Peer review of "Approaches to Dose Finding in Neonates, Illustrating the Variability between Neonatal Drug Development Programs"

_pharmaceutics, 2020, doi:10.3390/pharmaceutics12070685_

Round 1
Reviewer 1 Report
The paper from John van den Anker et al. provides an interesting insight into dose finding approaches for neonatal drug therapy by describing three different scenarios and at least one example for each of them. The authors are quite well known and experienced experts in neonatal drug therapy. The paper summarizes published research and related comments in a new forms. It is new and unique therefore. There are only a very few issues to be resolved:
- The paper is a little bit unbalanced concerning the discussed regulatory territory (US vs. EU), e.g. l.59: insert citation of the EU Paediatric Regulation; 4.1/l. 416 ff.: Peyona (higher dose) is authorized within Europe, the authorization of Gencebok is still pending in 2020: www.ema.europa.eu/en/medicines/human/summaries-opinion/gencebok
- You mentioned in silico methods and PK/PD studies as an example. A reflection on physiology-based PK models (PBPK) is provided in the last sentence only. However, at least for caffeine (Ginsberg et al., J. Tox. Environ. Health Part A, 2004) and Clopidogrel (Shebley et al., J. Clin. Ther., 2017) there is some literature available.
- Please add comments on the dosage forms for neonates which are used in your examples. The oral dosing of thyroid hormones is very small for instance. Are we sure that the neonate receives the correct dose?
Author Response
The paper from John van den Anker et al. provides an interesting insight into dose finding approaches for neonatal drug therapy by describing three different scenarios and at least one example for each of them. The authors are quite well known and experienced experts in neonatal drug therapy. The paper summarizes published research and related comments in a new forms. It is new and unique therefore. There are only a very few issues to be resolved:
We thank this reviewer for the very positive assessment of the paper.
- The paper is a little bit unbalanced concerning the discussed regulatory territory (US vs. EU), e.g. l.59: insert citation of the EU Paediatric Regulation; 4.1/l. 416 ff.: Peyona (higher dose) is authorized within Europe, the authorization of Gencebok is still pending in 2020: ema.europa.eu/en/medicines/human/summaries-opinion/gencebok
We agree and confirm that we have indeed exclusively described the drug development program as conducted within the United States and as assessed by the FDA in the examples. We have added this at the end of the introduction. However, we obviously also refer to the EMA regulatory territory for meropenem, and have added the content on caffeine, as suggested by the reviewer. This includes the different formulation (in the meanwhile, Gencebok has been authorized, 25 June 2020).
We felt that that the second alinea (including line 59) was already well balanced, as we aimed to focus on neonates, as the Pediatric Regulation was explicitly mentioned. Upon request of the reviewer, we have added the website link to the EU Pediatric Regulation.
- You mentioned in silico methods and PK/PD studies as an example. A reflection on physiology-based PK models (PBPK) is provided in the last sentence only. However, at least for caffeine (Ginsberg et al., J. Tox. Environ. Health Part A, 2004) and Clopidogrel (Shebley et al., J. Clin. , 2017) there is some literature available.
We fully agree as PBPK models for sure will provide a relevant tool to merge knowledge for initial dosing in neonatal drug development. We have added both examples to this alinea. However, both examples also illustrate the limitations and potential as only the caffeine PBPK example covers neonates. We have added this to the last alinea of the discussion: For caffeine, PK data as collected during clinical studies in neonates were used to further develop such PBPK models and to explore their performance [Ginsberg et al, J Toxicol Environ Health A 2004]. “In contrast, such PBPK models were also reported for meropenem and clopidogrel, but have not yet been extended to neonates [Verscheijden et al, PLoS Comput Biol 2019; Shebley et al, Clin Pharmacol Ther 2017]. This illustrates both the potential, but also the still existing opportunities to leverage these sources of information to further optimize dose selection for neonates”.
- Please add comments on the dosage forms for neonates which are used in your examples. The oral dosing of thyroid hormones is very small for instance. Are we sure that the neonate receives the correct dose?
We assume that the reviewer refers to formulations tailored to neonates to attain accurate (dose accuracy) and safe (excipient) drug exposure. This was not the topic of this paper, but is an important and valuable comment. We therefore have added this topic to the paper (early in the introduction, discussion) with a reference (making medicines baby size, O’Brien et al) recently published in an MDPI journal as open access to facilitate cross reading for the interested reader. In the discussion, we have also further reflected on this formulation issue for the drugs used as examples.
Reviewer 2 Report
Thank you for the oppertunity to review.
The paper highlights the complexity in drug development in neonates by illustrating different approaches for setting the dose of a drug substance, when applied in treatments of diseases either similar or related to that in to pediatric patients and adults, or unique to neonates.
Major comments:
The general introduction is not new and has been well decided elsewhere. Its suggested that this part is shortened substantially (line 49 to 119). Further, that the first section part 2 (line 130-136), which mainly apply to US regulatory guidance, is included in this general section, in an abbreviated version. The decision tree described is naturally followed by the three bullet points (line 120-127).
The first section in part 2 could be replace with line 140-147. As efficacy of antimicrobial treatment depends on the achievement of therapeutic concentrations at the infection site and this is challenging due to the altered drug disposition – this could have been presented in a structured manor, rather than details on off-label use, which would not be of particular relevance when guiding dose setting in drug development.
It should be considered that the chosen disease necrotizing enterocolitis is a medical condition in newborns and may therefore not be the most obvious example of a disease similarly treated in older pediatric patients or adults. The section (line 148-224) include a number of less relevant details and could be sharpened by including data in a structured manor across the various studies, rather than presenting a long summary of each study. Especially focusing on what AUC is desirable, how is this reached in adults and pediatric patients and what information is needed to tailor this exposure level to neonates, independently on the infection treated.
This section also shift focuses e.g. to include a section comparing various treatment regiments, which blurs the messages and should be left out (line 197-99, and line 197-199. It is also unclear whether the drug concentration leading to seizures was somewhat different in adults and pediatric patients compared to neonates. Nor is it discussed if these patients were receiving concomitant medications that could have contributed to seizures (e.g. by decreasing valproate exposure ect).
In section 3.1 and 3.2 Clopidogrel and Levothyroxine are chosen as an example of drugs in diseases related but not similar to adults. Its not clear how this influence dosing guidance in general. Again, extensive details are presented for neonates upfront – but is suggested to be presented in a structured manor across studies in adults and pediatric patients, and how this could be tailored to neonates and highlight the GAPs of particular interest, e.g. the CYP maturation and from what age e.g. polymorphism becomes relevant (as discussed in line 285-300 for clopidogrel), baseline conditions such as platelet aggregation differs (line 311-12), protein binding in case of levothyroxine (line358-362) and maturation of the HPT axis (line 406-412) ect – to be able to apply these GAPs to other disease areas. Preferably presented in a Table format in addition to the text. Its not before line 301-5 it become clear what the target exposure should be for clopidogrel and how it differs in adults and neonates and not mentioned at all in section 3.2 for levothyroxine. Nor ,how data from juvenile studies could be tailored in complex modelling and equally important when these data are not applicable, simply because MoA or off target is to different. Moreover, disease related covariates related to critical illness could be mentioned.
In section 4.1 and 4.2 Caffeine and lucintant are used as examples of disease only present in neonates. Caffeine is approved in both US and Europe, and the data presented here is more or less similar to the text in the SmPC, and could be shortened or left out. It could have been of interest to show in a structured manor how data can be compiled from various sources and elaborate more on the juvenile studies and how the initial dosages is set in neonates based on these data– where are the particular GAPS , how can we setup trials afterwards to learn more ect.
The discussion section is more a summary of the sections above and should be placed next to each section of relevance.
Author Response
Thank you for the opportunity to review.
The paper highlights the complexity in drug development in neonates by illustrating different approaches for setting the dose of a drug substance, when applied in treatments of diseases either similar or related to that in to pediatric patients and adults, or unique to neonates.
We thank the reviewer for the assessment of our paper, and comments provided and confirm that this was and is indeed the overall aim of the paper.
Major comments:
The general introduction is not new and has been well decided elsewhere. Its suggested that this part is shortened substantially (line 49 to 119). Further, that the first section part 2 (line 130-136), which mainly apply to US regulatory guidance, is included in this general section, in an abbreviated version. The decision tree described is naturally followed by the three bullet points (line 120-127).
There are some conflicting opinions between the reviewers, as the first reviewer asked to further extend this alinea and requested to add the European Regulation as a reference.
In line with reviewer 3, we removed the first sentence. We have removed the first section part 2 (former line 130-136) and provided this information in the introduction in a much shorter version. This has resulted in significant shortening. Howeever, the alinea from (first version), line 65 onwards reflects the ‘contrast’ between the legal setting and the subsequent ‘impact’ or ‘absence of impact’ of neonatal pharmacotherapy and labelling, and we feel that this information is crucial, relevant and not extensively described elsewhere.
The first section in part 2 could be replace with line 140-147. As efficacy of antimicrobial treatment depends on the achievement of therapeutic concentrations at the infection site and this is challenging due to the altered drug disposition – this could have been presented in a structured manor, rather than details on off-label use, which would not be of particular relevance when guiding dose setting in drug development.
As suggested, we have replaced the first section of part 2 by the (former) lines 140-147.
We agree on the reflections of the reviewer that efficacy of antibiotics depends on achievement of therapeutic targets at the site of infection, but that’s the very reason that we feel that the observation on variation in practices matters, because off label use explains this variation. We therefore have rephrased this sentence, but kept the alinea in the paper.
It should be considered that the chosen disease necrotizing enterocolitis is a medical condition in newborns and may therefore not be the most obvious example of a disease similarly treated in older pediatric patients or adults.
We agree, the disease is similar, not equal. This has been added and further stressed as the formal indication is indeed complicated abdominal infection.
The section (line 148-224) include a number of less relevant details and could be sharpened by including data in a structured manor across the various studies, rather than presenting a long summary of each study. Especially focusing on what AUC is desirable, how is this reached in adults and pediatric patients and what information is needed to tailor this exposure level to neonates, independently on the infection treated.
We have shortened this part of the paper significantly (1250 to about 920 words, so >-25%)
We hereby wanted to respect the suggestion of the reviewer, while we still wanted to maintain the narrative and the ‘history’. Besides efficacy (assessed by exposure), safety assessment remains crucial in this setting. Along the same line, we wanted to ‘keep’ the balance between US and European conducted studies, as suggested by all reviewers involved, so we have kept some data on the neoMero studies in the paper.
This section also shift focuses e.g. to include a section comparing various treatment regiments, which blurs the messages and should be left out (line 197-99, and line 197-199). It is also unclear whether the drug concentration leading to seizures was somewhat different in adults and pediatric patients compared to neonates. Nor is it discussed if these patients were receiving concomitant medications that could have contributed to seizures (e.g. by decreasing valproate exposure etc).
As requested, we have removed the former lines 197-199.
Although we are aware of the reported interaction in adults, valproate is an off target comment in neonates, as this is not used as first, second or even 3rd line AED in neonates. The assessment of seizures as a specific area of interest in these studies was mainly based on at that time available information in other non-neonatal populations (SmPC), but this turned out to be negative, or reassuring in neonates. As mentioned earlier, besides efficacy (assessed by exposure), safety assessments remains crucial in this setting. Along the same line, we wanted to ‘keep’ the balance between US and European conducted studies, as suggested by all reviewers involved, so we have kept some data on the neoMero studies in the paper.
In section 3.1 and 3.2 Clopidogrel and Levothyroxine are chosen as an example of drugs in diseases related but not similar to adults. Its not clear how this influence dosing guidance in general. Its not before line 301-5 it become clear what the target exposure should be for clopidogrel and how it differs in adults and neonates and not mentioned at all in section 3.2 for levothyroxine.
We have elaborated on this comment, as juvenile studies were crucial to develop a target exposure range (both efficacy and safety) for levothyroxine while the clopidogrel example has its historical merits, initiated by the written request and supported by in vitro data; We have added an introduction alineajust before the 3.1 and 3.2 section to provide this framework to the readership.
Again, extensive details are presented for neonates upfront – but is suggested to be presented in a structured manor across studies in adults and pediatric patients, and how this could be tailored to neonates and highlight the GAPs of particular interest, e.g. the CYP maturation and from what age e.g. polymorphism becomes relevant (as discussed in line 285-300 for clopidogrel), baseline conditions such as platelet aggregation differs (line 311-12), protein binding in case of levothyroxine (line358-362) and maturation of the HPT axis (line 406-412) ect – to be able to apply these GAPs to other disease areas. Preferably presented in a Table format in addition to the text.
We mainly wanted to use the clopidogrel as a case on how errors in dose determination can occur. To further stress this, we have added to the introduction a sentence just before sections 3.1. and 3.2 and have made a ‘separated’ alinea to further stress this in section 3.1. This means that the historical sequence and the evolving data and insights matter. We agree that this results in a somewhat more ‘narrative’ (less structured) section, but historically correct. Conversion to from what’s known in adults and children and tables will hamper this historically sequence so we really prefer to stick to the current reporting.
Nor ,how data from juvenile studies could be tailored in complex modelling and equally important when these data are not applicable, simply because MoA or off target is to different.
We have added a sentence on this aspect in the new introduction alinea just before sections 3.1 and 3.2 and have adapted the same sentence in section 3.2 to stress the ‘congenital hypothyroidism’ model.
Moreover, disease related covariates related to critical illness could be mentioned.
We assume that the author refers to critical illness hypothyroxinaemia, but this is still a very controversial topic, and not limited to neonates, with still ongoing academic discussion on whether this is an adaptive response or contributes to progressive disease (eg Maiden et al, Crit Care Clin 2019). We refer to the already existing alinea (Besides congenital hypothyroidism, postnatal thyroid hormones in preterm neonates have also been evaluated to prevent morbidity and mortality and in preterm cases with transient hypothyroxinemia. Based on meta-analysis for both indications (primary or secondary prevention), there is insufficient evidence for any of these indications [45]) and suggest not to further elaborate on this aspect.
In section 4.1 and 4.2 Caffeine and lucintant are used as examples of disease only present in neonates. Caffeine is approved in both US and Europe, and the data presented here is more or less similar to the text in the SmPC, and could be shortened or left out. It could have been of interest to show in a structured manor how data can be compiled from various sources and elaborate more on the juvenile studies and how the initial dosages is set in neonates based on these data– where are the particular GAPS , how can we setup trials afterwards to learn more ect.
We understand this suggestion, but we preferred to stick to the ‘historical’ sequence for the caffeine, as the CAFCIT trial does illustrate how a single study can be pivotal for registration. As requested by the first reviewer, we have added the European setting. The use of the caffeine data to explore or ‘validate’ PBPK models has been added in the final alinea of the discussion.
The discussion section is more a summary of the sections above and should be placed next to each section of relevance.
We agree that this is to a certain extent a ‘summary’, but it also provides and repeats the overview and the variability in drug development programs. We have carefully considered this suggestion, but this is somewhat in contrast to the suggestions of the other reviewers (eg PBPK extension in the discussion section) and the adaptation of the first alinea of the discussion section. The revised version contains more information and reflections on the PBPK aspect, as well as the formulation issue.
Reviewer 3 Report
This is a detailed and thorough review on drug development programmes for neonates, using various clinical scenarios to highlight the key processes involved.
Major issues:
This is a well structured review, with immense detail for each drug covered. The only real limitation I can find is that there are multiple ways of deriving the data for improved dose in neonates, and the authors use different drugs to exemplify this - but there could be a little more signposting near the beginning that the main text will say how the different method have been used, and it is the discussion that will detail the overall pros and cons of these methods in neonates.
Minor Issues:
The second sentence is a non-sequitur from the first sentence. Either lose the entire first sentence, or adapt to mention the need for neonatal specificity.
Line 57-58 - "This Act also focused on one of the neglected pediatric groups in drug development, including neonates" - suggest amending "including" to "namely" or similar
Line 98 - PK abbreviation used and not explained previously (also PD on line 100)
Line 141 - amend "potential" to "a potentially"
Line 199 - Sentence starting "compared" - New paragraph perhaps?
Line 205 to 224 - safety section. Another justification for good quality safety studies is the very poor reporting of suspected ADRs to regulatory agencies.
Table 4 - the use of "not" in the final column - would suggest just say either prevention, treatment, or both
Author Response
This is a detailed and thorough review on drug development programmes for neonates, using various clinical scenarios to highlight the key processes involved.
Major issues:
This is a well structured review, with immense detail for each drug covered. The only real limitation I can find is that there are multiple ways of deriving the data for improved dose in neonates, and the authors use different drugs to exemplify this - but there could be a little more signposting near the beginning that the main text will say how the different method have been used, and it is the discussion that will detail the overall pros and cons of these methods in neonates.
We thank the reviewer for this very positive and supportive assessment of the paper. We indeed wanted to stress that there are multiple ways to derive data for improved dosing in neonates, be it that we wanted to stress the differences in strategies needed during the clinical studies (PK, PD, safety), and not so much how preclinical data or PBPK models can be used for dose selection. We have further extended the two last alineas of the introduction on this aspect, and have added a sentence in the first alinea of the discussion. Furthermore, we have adapted the title and have shifted the text in the abstract in an attempt to better stress our aim, i.e. to illustrate the variability between neonatal drug development programs.
Minor Issues:
The second sentence is a non-sequitur from the first sentence. Either lose the entire first sentence, or adapt to mention the need for neonatal specificity.
We have removed the first sentence
Line 57-58 - "This Act also focused on one of the neglected pediatric groups in drug development, including neonates" - suggest amending "including" to "namely" or similar
We have amended ‘including’ to ‘namely’
Line 98 - PK abbreviation used and not explained previously (also PD on line 100)
We have added these abbreviations in the preceding sentence
Line 141 - amend "potential" to "a potentially"
Amended
Line 199 - Sentence starting "compared" - New paragraph perhaps?
Amended
Line 205 to 224 - safety section. Another justification for good quality safety studies is the very poor reporting of suspected ADRs to regulatory agencies.
We have added this issue in the alinea, with a recent reference on this topic (Davis et al, J Pediatrics) as safety assessment and reporting is an issue in neonates throughout the life cycle of
Table 4 - the use of "not" in the final column - would suggest just say either prevention, treatment, or both
Amended
Round 2
Reviewer 2 Report
The Authors
have performed major revision of the article and its now well read